# How Did the COVID-19 Pandemic and Digital Divide Impact Ciganos/Roma School Pathways?

Susana Mourão [1,2], Sara Pinheiro [2,3], Maria Manuela Mendes [2,4,*], Pedro Caetano [5] and Olga Magano [2,6]

1 Centro de Investigação em Psicologia (CIP-UAL), Universidade Autónoma de Lisboa, 1169-023 Lisboa, Portugal
2 Centro de Investigação e Estudos de Sociologia do Iscte-Instituto Universitário de Lisboa (CIES-Iscte), 1649-026 Lisboa, Portugal
3 Centro de Investigação e Intervenção Educativas (CIIE), Universidade do Porto, 4200-135 Porto, Portugal
4 Instituto Superior de Ciências Sociais e Políticas, Universidade de Lisboa, 1300-663 Lisboa, Portugal
5 Centro Interdisciplinar de Ciências Sociais, Universidade Nova de Lisboa, 1099-085 Lisboa, Portugal
6 Departamento de Ciências Sociais e de Gestão, Universidade Aberta, 1269-001 Lisboa, Portugal
* Correspondence: mamendesster@gmail.com

**Abstract:** The COVID-19 pandemic forced the Portuguese government to declare various lockdowns between 2020 and 2022. The first State of Emergency was enforced in March 2020, in which face-to-face classroom teaching was repeatedly interrupted. At that time, families were expected to provide the necessary supplies for digital learning, with some support from the government, municipalities, civil society, and local institutions. Nevertheless, many families already lived under precarious conditions before the pandemic, and so the lockdown measures increased their vulnerability, with a probable impact on student school attendance and conditions enabling academic success. Since Ciganos/Roma are part of this vulnerable population, we intend to explore how the COVID-19 pandemic impacts the school pathways of these students, namely in secondary education, where they represent a minority group. The data are derived from a variety of qualitative sources collected during research carried out in the two Metropolitan Areas in Portugal. The COVID-19 pandemic affected the youngsters' access to classes and their motivation to attend school, and opens the discussion about how because of the government's universal measures, by failing to consider social diversity, in particular Ciganos/Roma Ciganos/Roma families, this pandemic crisis may disproportionally affect the education of their children and youth. The findings highlight, firstly, that these impacts continue to be rendered invisible and naturalized in the public sphere and, secondly, that the measures and legislation underlying the pandemic effects continue not to include Ciganos in policymaking processes.

**Keywords:** Ciganos/Roma; COVID-19 pandemic; school pathways; digital divide; Portugal

## 1. Introduction

This article emerged due to the importance of debating the impacts of the COVID-19 pandemic, initially breaking out in different countries worldwide in late 2019 and in Portugal in early 2020, leading to the first lockdown in March 2020, followed by various other lockdown periods throughout 2020–2022. In March 2020, in light of the rise in the number of people infected with COVID-19, the Portuguese government was forced to declare the first State of Emergency and, consequently, decree the interruption of face-to-face classes (as well as other school and extracurricular activities) for a period of 6 weeks (Presidência do Conselho de Ministros 2020a, 2020b). Hence, and with the support of government entities, families were required to readjust, to be ready, willing, and able, and provide the necessary technological devices to secure access to distance/digital learning for their children at school. The children of families experiencing greater social and economic difficulties continued to have guaranteed meals at school. However, and despite all the State efforts, it should be stressed that this situation significantly contributed to

enhancing the pre-existing social, economic, and cultural inequalities, with an impact on the educational pathways of more financially vulnerable students, which is the case of the Ciganos[1]/Portuguese Roma youth and families followed under the EduCig Project[2].

Similar to what happened in Portugal, all the other governments of European countries imposed a diversity of measures as the pandemic spread throughout Europe from 2020 onwards, disproportionally affecting people in situations of greater social and economic vulnerability, such as the Roma/Ciganos and other ethnic minorities (FRA 2020).

The various lockdown periods not only required families to adjust their work and routines to supervise their children of school age, through tele-learning and online classes, but it also created the expectation that they would manage to provide the necessary material for distance/digital learning, in particular the digital devices, with the economic support of the government, municipalities, civil society, and local institutions. While it is certain that many Ciganos/Roma families were already in a socially and economically vulnerable position and living under precarious conditions before the pandemic, the lockdown measures acted to increase social inequalities (FRA 2020; Korunovska and Jovanovic 2020). This was especially because, in the Portuguese context, a large number of Ciganos/Roma households are economically dependent on fairs and markets where they are vendors, and with the closure of these venues, they were plunged into a situation of enhanced economic fragility, and consequently increased social and cultural vulnerability, due to being prevented from earning income that was not covered by the economic support given by the State to companies and other workers[3].

Concerning education, Ciganos/Roma children and youngsters show difficulties in access, or even absence of access, to technological equipment (i.e., computers) and the internet, combined with enormous hardship in supervising distance learning (FRA 2020). Moreover, it was observed that there were parents and guardians with no formal education or insufficient schooling to enable them to supervise homeschooling, as well as an impossibility of supporting local projects to that end (Lopes 2021).

It is thus understandable that the pandemic and its successive lockdowns should have exacerbated the existing digital and socioeconomic divide between people and families living in more precarious socioeconomic conditions (Murat and Bonacini 2020), and hence, the relevance of this article, that will enable considering the impacts on the access and performance of young Ciganos/Roma in secondary education in Portugal. Even more negative impacts are expected, caused by the effects of the pandemic, which will be discussed throughout this article.

This paper aims to explore how the COVID-19 pandemic impacts the school pathways of Ciganos/Roma students in secondary education. We begin by presenting the theoretical framework for factors that facilitate and/or hinder successful educational pathways of Portuguese Roma/Ciganos youngsters in secondary education. This is followed by a discussion of the methods used that mobilize triangulated data from different qualitative sources of information that represent key roles in fostering better educational achievements among Ciganos/Roma students (Abajo and Carrasco 2004; Gamella 2011; Gofka 2016; Magano and Mendes 2021a), namely (i) interviews with professionals from community-based projects and Evangelical pastors; (ii) focus groups including Roma/Ciganos students, their families, Roma/Ciganos activists, intercultural mediators, social workers, and teachers; and (iii) ethnographic analysis of two young students' daily lives. Finally, a reflexive and comprehensive analysis is given of the findings and discussion, with final considerations.

## 2. Contextual Background

Since the outbreak of the COVID-19 pandemic, there has been frequent news and debate on the social networks in which Ciganos/Roma are accused of spreading the disease, and a form of moral panic (Berescu et al. 2021) was generated, holding them responsible for the contamination of their fellow citizens somewhat all over central and southern

Europe (Plainer 2020; Costache 2020; Berescu et al. 2021), fostering feelings of hatred and xenophobia, kindling movements of discrimination.

In general, in Europe, the Ciganos/Roma were especially vulnerable to COVID-19, "they have been disproportionately affected by the disease, both directly (in terms of increased rates of infections, hospitalisations, and deaths, although official data for the latter are unavailable) and indirectly (e.g., in terms of increased inequality and stigmatisation)" (Holt 2021). We argue that pre-existing structural inequalities and racialized vulnerability, as well as antigypsyism and hate narratives against Ciganos and Roma, were reinforced during and post-COVID-19. Accordingly, and for a deconstruction of stereotypes, we are interested in understanding how the measures implemented in Portugal, during that time, attempted to combat the existing structural inequalities, in relation to Ciganos/Roma families.

Pre-pandemic structural problems experienced by some Ciganos/Roma, such as the absence of decent housing conditions, difficulty in obtaining means for ensuring basic hygiene, economic hardship due to lack of employment, and/or job insecurity worsening (Mendes et al. 2014). Added to these problems is the need to produce more meals at home plus the extra expenses related to health protection and disinfection materials, leading to higher economic costs for families and contributing to the increased exposure of children and youth to the pandemic (Mendes 2020), with the difficulty in being supported by institutions and local intervention projects also being revealed (Lopes 2021).

In addition to food and other basic goods and services, it was expected that Ciganos/Roma families should have the conditions to provide the necessary learning environment (computer, internet, a place to study at home) to secure distance/digital learning albeit with some support from the government, municipalities, civil society, and local institutions. However, some families were already living under precarious conditions before the pandemic (Mendes et al. 2014); hence, the lockdown measures increased their difficulties in surviving, with a likely impact on the school attendance and achievement of these students. It is important to stress that these factors were further exacerbated by the fact that the majority of Ciganos/Roma parents/guardians have very low schooling levels and/or are even illiterate (Mendes et al. 2014), indicating that they do not have the digital knowledge required to supervise their children of school age, and often do not have access to computerized resources (Medinas and Magano 2020).

It is interesting to debate the government's universal measures to combat the pandemic, which did not consider the social diversity and socioeconomic conditions of Ciganos/Roma families, disproportionally affecting the education of their young students (Magano and Mendes 2021b) in that period, when compared with other children and youth of families with the necessary economic resources and skills for their educational supervision.

### 2.1. Educational Situation before the Pandemic

The findings of the National Study on Ciganos/Roma Communities (Mendes et al. 2014) demonstrate that the use of a computer and access to the internet were only available to 17% of the interviewed[4], with 27.5% having stated that they lived in shacks, rudimentary, or wooden houses (Mendes et al. 2014). More recently, the European Union (EU) Agency for Fundamental Rights (FRA 2022) stressed that 80% of the Ciganos/Roma population in the EU live below the poverty threshold of their country, whereas that figure is 96% in Portugal, adding that the risk of child poverty among Ciganos/Roma stands at 97%. The pre-pandemic educational situation of the Ciganos/Roma was already very bleak. Indeed, they were among those primarily affected by educational inequalities, as the majority continue to have low schooling levels and high rates of educational underachievement and school dropout. Various vulnerabilities are persistent and superimposed among Ciganos/Roma, such as poverty, illiteracy, exclusion, and social discrimination (FRA 2012, 2018, 2019), who are also frequently subjected to attitudes of segregation and racism, also due to migratory movements in Western Europe (Powell and Lever 2017; Crețan et al. 2021), inclusively in a school context (FRA 2018, 2019, 2022). Central and Eastern European countries, such as

the Czech Republic, where statistical data showed, in 2007, that Roma/Ciganos "were 27 times more likely to be educated in special schools for students with mental disabilities", believing that any discrimination could thus be justified (O'Nions 2010, p. 1). In Hungary, since 1989, the school segregation of Roma/Ciganos children has been increasing, where Roma/Ciganos children continue to be separated from their non-Roma peers (Messing 2017). From this point of view, it can be stated that we are facing complex segregation processes in mainstream education (Messing 2017), which privilege excluding those who are seen as different and not embracing integration processes. A qualitative study conducted in the Lisbon Metropolitan Area (Mendes 2007) found some situations and contexts of discrimination predominantly configured the following: (a) conduct and practices in terms of differential (and unfavorable) treatment by teachers and staff; (b) practices of segregation between schools or within a given school; (c) forms of verbal discrimination, whether positive or negative, from teachers, staff, and peers. Intercultural education needs to overcome, through integration, any difference, so that the success of educational initiatives is not measured only by academic performance (O'Nions 2010). It is true that in recent years there has been an increment of public policies in the national and European Community sphere aimed at fighting against inequality and breaking vicious cycles of exclusion that are perpetuated from one generation to the next. Nevertheless, these public policies have been insufficient to overcome the mechanisms and processes historically rooted in social structures that produce and reproduce structural and systemic inequalities reflected in various spheres of the life of Ciganos/Roma persons. Therefore, although we are currently witnessing the most highly educated Roma/Ciganos generation ever, we are still faced with a high number of cases of educational underachievement and early school leaving, bringing about serious implications in the fight against marginality (Crețan and Turnock 2009), since it is argued that education is the fundamental tool for integration into society.

When analyzing the national statistics on the schooling level of the Ciganos/Roma population, a disturbing question is immediately raised linked to the notorious underrepresentation of their students in secondary education in Portugal. In the academic year 2018/19, only 2.6% of young Ciganos/Roma were enrolled in secondary education (DGEEC 2018/2019). Previous studies conducted in Portugal of qualitative nature (Magano and Mendes 2021a; Mendes 2020; Magano and Mendes 2016) provided evidence for these numerical data, demonstrating that the number of young Ciganos/Roma who completed secondary education is still very low. The statistics are not particularly promising at a European level, as, in 2016, 63% of Roma aged between 16 and 24 years old were not in education, employment, or training (NEET youth); which contrasts with the EU average of 12%, for the same age cohort (the 2019 data are very similar). A recent FRA report (2022) shows that the rate of 20 to 24-year-old Ciganos/Roma who have attained at least upper secondary education is 27% across all EU countries. This figure is the lowest in Portugal, Greece, Czechia, and Romania (between 10% and 22%). Greece and Portugal are also the countries with the largest differences in rates between Ciganos/Roma and the general population.

National and international studies in pre-pandemic scenarios reveal various important factors for the educational achievement or underachievement of Ciganos/Roma children and youngsters. At a national level, the key factors identified are family support and reference models but also friendly and inclusive school conditions (Mendes and Magano 2016; Magano and Mendes 2016; Magano and Mendes 2021a) inclusively as a result of the impact of public policies fostering social protection and education in the life of Ciganos/Roma families (Magano and Mendes 2014).

## 2.2. Determinant Factors in Educational Achievement and/or Continuity

The factors constraining and facilitating the educational continuity pathways of these youngsters could be numerous, dynamic, and interrelated. These motivations and factors have been studied throughout the last two decades in a European context, especially by (Abajo and Carrasco 2004; Bereményi and Carrasco 2017; Brüggemann 2014; Gamella 2011;

and Gofka 2016). Overall, their models and propositions argued that the Roma/Ciganos educational achievement and continuity can be attained in a variety of personal and social circumstances, such as positive valuation of academic potentialities, not only individual, but also related to support from family, school, local and wider social community; and construction of a project of educational continuity, sustained by personal skills, economic resources, and significant reference models (Figure 1). Hence, this research reinforces, on the one hand, questions of socioeconomic differentiation and the need to consider the relationship between various explanatory factors in analyzing successful pathways. On the other hand, and above all due to taking into account the perspective of the actual protagonists, it emphasizes the individual's competences, skills, and agency capacity (and resilience) along these pathways, in the meaning developed by Giddens (2004).

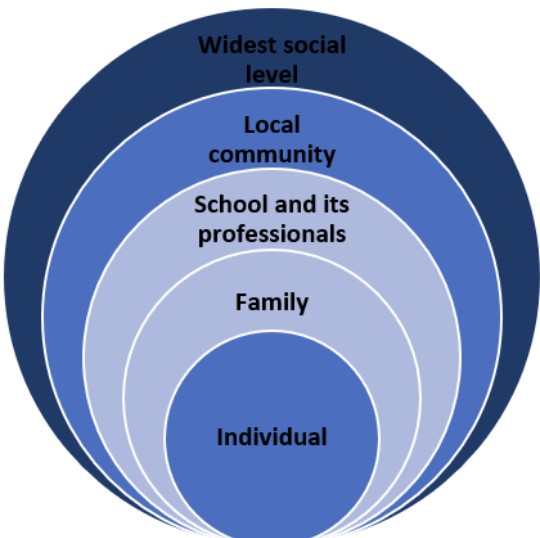

**Figure 1.** Interpretative model by Gamella (2011). Source: EduCig Project, 2021.

In Portugal, between 2014 and 2015, in a study conducted on the individual/family and education relationship, it was found that teachers and technical staff hold the Ciganos/Roma students and their families accountable for lack of interest in school, educational under-achievement and early school leaving, without ever critically questioning the school organization, its program, methodologies, and other teaching and learning frameworks. Nevertheless, in all truth, it must be said that the implemented educational policy measures (PIEF[5], TEIP[6], Novas Oportunidades (New Opportunities)[7]) were almost always forms of agglutinating, segregating the students into specific year groups.

In sum, we can state that it is in the multiplicity of interactive and dynamic factors that we find the opportunities for educational access, continuity, and achievement, with points of confluence being established that directly interfere with those opportunities for educational access and achievement. Thus, in the particular stressful context of the pandemic and associated lockdowns, where online learning is an unavoidable day-to-day reality, becomes imperative to understand how the abovementioned dynamics and interrelated determinants will operate to impact (or not) the school pathways of Ciganos/Roma students in secondary education; and especially since is already identified some of their previous socioeconomic and digital inequalities.

## 3. Materials and Methods

The presented data come from a broader funded project, which, overall, aims to understand the pathways of Ciganos/Roma students attending secondary education in the Metropolitan Areas of Lisbon and Porto, and their aspirations for access to higher education. Qualitative data collected during the pandemic and especially throughout the first lockdown in Portugal (6 weeks in 2020, March to May, when classroom lessons and

other extracurricular activities were forcibly shut down) were mobilized. As detailed in the Introduction (Section 1), these sources of information include heterogeneous key actors with relevant roles in fostering better educational achievements among Ciganos/Roma students (Abajo and Carrasco 2004; Gamella 2011; Gofka 2016; Magano and Mendes 2021a) and will be detailed below.

### 3.1. Qualitative Data Collection with Different-Level Participants

#### 3.1.1. Interviews with Professionals from Community Projects and Evangelical Priors

Particularly during the pandemic, 8 professionals from 6 Escolhas Projects in Northern Portugal were interviewed, the majority located in the Porto Metropolitan Area. It was intended to gain a better understanding of the impact of the local community, peer networks, and participation in the Portuguese governmental Escolhas Program[8] on the school pathways of Portuguese Ciganos/Roma students. These interviews were held alongside those conducted with the young students (total of 31), as the latter gave us important clues to develop the semi-structured interview scripts for the professionals. It included questions about the main characteristics of interviewed people and their projects, and about the context of Ciganos/Roma children, young people, and families who are engaged in the project, especially their educational pathways.

The projects' key objectives and activities were sustained by three major areas: (1) promotion of educational achievement and prevention of dropout and absenteeism; (2) employment, training, and entrepreneurship; (3) associative/community dynamics, participation, and citizenship. Professionals, whose sociodemographic and professional characteristics are detailed in Table 1, mostly performed coordinating tasks.

**Table 1.** Sociodemographic and professional characteristics of professionals in the Escolhas Projects.

| Interviewee | Sex | Age | Schooling Level | Position |
|---|---|---|---|---|
| E1 | Female | 39 | Bachelor in Social Service | Coordinator |
| E2 | Male | 45 | Bachelor in History of Art and Master in Artistic Studies | Coordinator |
| E3 | Male | 49 | Bachelor in Communication Sciences | Coordinator |
| E4 | Female | 29 | Bachelor in Social Service | Coordinator |
| E5 | Female | 32 | Bachelor and Master in Education Sciences | Coordinator |
| E6 | Female | 31 | Master in Education | Coordinator |
| E7 | Female | 21 | Bachelor in Social Education | Technician |
| E8 | Male | 38 | Master in Education | Technician |

In order to analyze another important key factor in school continuity among young Ciganos/Roma, namely religion (Gamella 2011; Mendes et al. 2014), 3 Evangelical pastors from the Lisbon Metropolitan Area were interviewed, whose church followers included Ciganos/Roma, especially young students. The developed semi-structured script comprised questions about the interviewees' personal context (residency and mobility, family, work), characteristics and history of their church, information about Ciganos/Roma attending the church, and their representations about the relationships between religion, education, and discrimination. Because all these interviews were held exclusively during the pandemic period, the script also had specific questions about the impact of COVID-19 on the daily church routine.

Evangelical pastors were all male; aged 38, 43, and 46 years old; and lived in municipalities near Lisbon. Two of them were Ciganos/Roma, both parents also, and the other was Angolan. They had basic schooling, respectively 6, 7, and 9 years, with one of them, at the time of the research, attending an adult model course to achieve the 3rd cycle (9 years of school). All the interviewees had professional activities alongside their church responsibilities, two of them in a more informal manner (fair vending and temporary civil construction work), while another recently started to work at a private transport company. They were all married to Ciganos/Roma, all with a lower educational level than their

husbands. All had children (between two and four), the majority being students; only one girl had dropped out of secondary education.

### 3.1.2. Focus Groups with Ciganos/Roma Students, Their Families, Evangelical Priors, Activists/Associative Members, and Intercultural Mediators

In order to more easily access consensual and contrasting perspectives about the key factors that influence young Ciganos/Roma to study in secondary education, we conducted three focus groups (Krueger and Casey 2000; Morgan 1988), with different relevant actors for their educational achievement: (1) Ciganos/Roma students at secondary school and university; (2) Ciganos/Roma parents, intercultural mediators, Ciganos/Roma activists and associative members with experience in school intervention, and Evangelical pastors; (3) professionals with work experience with Ciganos/Roma students. We tried to diversify participants as much as possible in terms of their sociodemographic and professional characteristics, and in Focus Group 2, some of the participants simultaneously had more than one role or belonging. They were recruited by a snowball sampling approach and using the team's previous relational networks.

The areas proposed for discussion in each focus group resulted from the interviews' main findings, theoretically supported by the abovementioned model proposed by Gamella (2011). Accordingly, the developed semi-structured script aimed to access the participants' perceptions regarding social and educational pathways of Ciganos/Roma, determinants of Ciganos/Roma educational performance and the school's role in their evaluation process, relationships between school and Ciganos/Roma families, and information about pedagogical curriculum and methodologies adopted.

Ciganos/Roma students from Focus Group 1, two boys and one girl, were aged between 19 and 20 years old. They come from different geographic regions: Lisbon Metropolitan Area, Porto Metropolitan Area, and Central Portugal. Two of them were at university (the girl taking a course in International Relations and the boy taking Law), and the other was a secondary student who had recently completed the regular course in Humanities and aspired to take a higher education course in Law or Communication. The female university student had completed secondary education following the regular course in Humanities and had her ambition set on a diplomatic career. The male university student had completed secondary education in a vocational course.

The 11 heterogeneous role participants from Focus Group 2 were 5 men and 6 women, aged between 20 and 55 years old. Two of them lived in the Porto Metropolitan Area and one in Central Portugal, with all the others being from the Lisbon Metropolitan Area. Table 2 details their sociodemographic and professional characteristics, being relevant to highlight that three of them were studying at the time of the research (two women at university). One other woman had also previously attended the university.

**Table 2.** Sociodemographic and professional characteristics of the Focus Group 2 participants.

| Participant | Sex | Profession or Status | Area of Residence |
|---|---|---|---|
| P1 | Male | Evangelical Pastor and Intercultural Mediator | Lisbon Metropolitan Area |
| P2 | Female | Student and Activist | Lisbon Metropolitan Area |
| P3 | Male | Fair Vendor, Activist | Porto Metropolitan Area |
| P4 | Female | Vice-chair of a Cigano Association, Activist, and Intercultural Mediator | Central Portugal |
| P5 | Male | Activist | Lisbon Metropolitan Area |
| P6 | Female | Vice-chair of a Cigano Association | Lisbon Metropolitan Area |
| P7 | Female | Activist | Porto Metropolitan Area |
| P8 | Female | Intercultural Mediator | Lisbon Metropolitan Area |
| P9 | Male | Evangelical Pastor | |
| P10 | Male | Evangelical Pastor | Porto Metropolitan Area |
| P11 | Female | Intercultural Mediator and Trainer | Lisbon Metropolitan Area |

The 10 heterogeneous role participants from Focus Group 3 were 3 men and 7 women, aged between 24 and 50 years old. Contrary to Focus Group 2, the majority of them lived in the Porto Metropolitan Area, in different municipalities, with only two being from the Lisbon Metropolitan Area. Not surprisingly, the majority had university studies, except one Ciganos/Roma intercultural mediator who had 9th-year schooling. Table 3 details the sociodemographic and professional characteristics of these participants.

**Table 3.** Sociodemographic and professional characteristics of Focus Group 3 participants.

| Participant | Sex | Schooling Level | Profession | Institution Represented |
|---|---|---|---|---|
| P1 | Female | Higher Education | Psychologist | School Grouping |
| P2 | Male | 9th Year | Intercultural Mediator | Association Supporting Social and Community Integration |
| P3 | Female | Higher Education | Service Technician | School Grouping |
| P4 | Female | Higher Education | Community Educator and Escolhas Project Coordinator | Sports Association for Social Change and Inclusion |
| P5 | Female | Higher Education | Social Educator | School directed at Prevention of Early School Leaving |
| P6 | Female | Higher Education | Higher Education Technician | NGO |
| P7 | Male | Higher Education | Participant in socioeducational project | NGO |
| P8 | Male | Higher Education | Teacher | Secondary School |
| P9 | Female | Higher Education | ——— | School Grouping |
| P10 | Female | Higher Education | Teacher | School Grouping |

3.1.3. Ethnographic Observation from Roma/Ciganos Young Students' Daily Life

In order to gain a more in-depth grasp of the social diversity and complexity found with the abovementioned methods, two of the young interviewees were also observed on a daily basis, using an ethnographic method, which enables drawing the researchers and participants closer together in their real contexts, also contributing to collaborative knowledge development (Setti 2017). The selected youth had previously established deeper relationships with the research team, essential to the ethnographic data collection, and also have different daily lives and family contexts. At the beginning, they were locally observed in their neighborhoods and residences, near the school and along their school–home routes. Since the pandemic, they were exclusively followed by remote devices, using video conference platforms, interactions in social media, written messages, and phone calls.

Just as in the youngsters' interview scripts, the ethnographic observation focused on different-level key factors that may influence Ciganos/Roma educational pathways (Gamella 2011; Gofka 2016; Magano and Mendes 2016), namely characteristics of household contexts and school–home routes, family and other significant relationships, identity construction, and study and learning habits. To maintain their spontaneous behaviors and discourses as much as possible, independently of the researcher's presence, the observations were systematically recorded in a field diary, but only at the end of each interaction.

The two Ciganos/Roma students who were ethnographically observed were both from the Lisbon Metropolitan Area, although from different municipalities, and were 17 years old. Neither lived in council housing. Differently from the majority of Ciganos/Roma students (Magano and Mendes 2016), they had both completed compulsory education in a regular course. However, at the research time, one of them (fictional name of João) was attending an Audiovisual vocational course, and the other (fictional name of Lucas) was taking a regular course in Humanities.

### 3.2. Data Analysis, Ethics, and Quality Criteria

Data transcribed from the audio-taped interviews and focus groups, and the written ethnographic field diaries, were analyzed through a bottom-up content analysis (Bardin 2011; Maroy 1997), to identify different-level impacts that the pandemic and the first lockdown period raised to Ciganos/Roma educational pathways. Data were firstly coded by the first paper's author, while the others accessed the main results to identify and resolve discrepancies in data interpretation. Mutual exclusivity among the created categories/subcategories was assured. All the procedure was supported by MaxQda software.

Participants were previously informed about the study's main objectives and methodologies, and gave their informed consent to participate. The project and its data collection instruments (namely the scripts) were also approved by the University's Ethics Commission.

## 4. Results and Discussion

### 4.1. Beyond the Visible Effects of the COVID-19 Pandemic on Roma/Ciganos School Pathways

Figure 2 frames the triangulated perception of the different participants about the most significant impacts of the pandemic on the educational pathways of Ciganos/Roma students in secondary education, showing a diversity of outlooks, organized into three axes of analysis: direct constraints in access and supervision of classes and school dynamics, indirect effects on student motivation, and a greater socioeconomic vulnerability, especially with respect to the socioeconomic difficulties and material conditions for scholastic purposes faced by these families. Identification of these different-level, but mutually interrelated, determinants of Ciganos/Roma educational achievement and/or continuity during the particular circumstances of COVID-19 reinforce the pertinence from already developed models on the overall context of the Ciganos/Roma education field (Abajo and Carrasco 2004; Bereményi and Carrasco 2017; Brüggemann 2014; Gamella 2011; Gofka 2016). Indeed, the factors constraining and facilitating Ciganos/Roma educational pathways specifically during the pandemic and digital learning also attained a variety of personal and social circumstances, where potential opposing forces between the families' socioeconomic conditions and the wider community or social level may particularly exacerbate several of their previously identified inequalities on school access and continuity. More specifically, and namely during the lockdown periods, being digitally excluded clearly impacts Ciganos/Roma student educational engagement, being also responsible for relevant disruptions in their significant social networks (Velicu et al. 2022). This social discontinuity, even indirectly, is also expressed as a relevant predisposing factor for Ciganos/Roma school success and continuity during the COVID-19 pandemic, making sense of how students' digital differentiation may differently impact their school pathways.

| Direct constraints in Access and Supervision of Classes and School Dynamics | Indirect effects on Student Motivation * | Greater Socioeconomic Vulnerability |
|---|---|---|
| • No digital resources (n = 21) <br> • Suspension and/or difficulty in readaptation to classes (n = 15) <br> • Inadequate space for studying (n = 11) * <br> • Change in assessment processes and risk of not passing the year (n = 10) <br> • Digital illiteracy of parents/family (n = 4) | • Abrupt cutting of social relations (n = 14) <br> • Suspension and/or difficulty in readaptation to religious/extracurricular activities and Escolhas projects (n = 12) <br> • Breaking of routines (n = 3) | • Interruption of work-related activities and difficulty in job search (n = 21) * <br> • Absence of social protection (n = 2) <br> • Increased family expenses due to lockdown (n = 11) ** |

**Figure 2.** Perception of the different-role participants about the impacts of the pandemic on educational pathways. Note: * Emphasized only by Ciganos/Roma Youngsters and Families/Activists. ** Emphasized only by Technical Staff of the Escolhas Projects.

### 4.1.1. Digital Divide and the Non-Access to Online Classes

As illustrated in Figure 2, it is evident that there are more visible and direct constraints in access to classes, above all related to limitations in access to digital equipment and services that are indispensable for decent supervision of distance/digital learning.

*"(…) in the Ciganos/Roma communities, not everyone had and has access to the same conditions to enable the sufficient number with access to computers, for example, in my case, in the early days there were only two computers, and I have four students (…) two at university (…) and the other two in secondary education. And it was very complicated, very complicated."* (Focus Group 2, Family member of Young Students)

*"(…) if the majority community was frightened, let's say, was affected, shall we say, by the pandemic, it is even more disturbing in the Ciganos/Roma community. So, there are people attending school online, but there are people who don't have a computer (…)"* (Focus Group 3, Teacher)

These perceptions, widely shared by Ciganos/Roma people and also by the community or school professionals, were in loco corroborated by the ethnographic follow-up of one of the young students in secondary education. Indeed, Lucas lives in an area where the optical fiber coverage is inadequate, making sense of how the quality of devices and internet access may directly influence the success of digital learning during COVID-19, namely from more disadvantaged students (Fejes and Szűcs 2021).

*Lucas' access to the internet could be problem for the resumption of online classes*, via *the Zoom platform. In his area, there is simply no internet by optical fiber, and the internet he receives at home is costly and slower, and all his neighbors complain.* (Field Diary, Ethnographies with Young Students, April 2021)

Particularly, the students and their parents also relied on the inability of Roma/Ciganos family members to support homeschooling, often due to certain digital illiteracy. Although less emphasized, this assumption reinforces the idea that Ciganos/Roma educational gaps during the pandemic go beyond government provision of digital supplies and so may also imply individualized training in using online/distance learning effectively (FRA 2020). If before the pandemic the community-led projects could have had an important role in minimizing Roma/Ciganos families' illiteracy, during the COVID-19 pandemic, they also reported to struggle with difficulties in the extracurricular activities and dynamics supporting homeschooling, requiring constant readaptation not always easy to achieve. Thus, the surrounding local community as a potential resource for Ciganos/Roma students' engagement at school is at this time also somehow weakened.

*"Now, at this stage of the pandemic, we are adapting and it's being more difficult, because there are many who don't have access to the online workshops, a great number in fact."* (Interview, Technician of the Escolhas Project, AMP)

It is important to highlight that these types of difficulties are reported across the board by the different actors who participated in the research, demonstrating that the universal measures decreed by the government might not be meeting the diversity and heterogeneity of needs of different families (whether Ciganos/Roma or not). Instead, they end up actually contributing to increasing the digital divide that already existed before the pandemic between students with more and less economic resources (Rideout and Katz 2016), with this divide having further intensified especially during the lockdown periods and consequent distance teaching (Bešter and Pirc 2020).

In addition to the difficulties of access to digital devices enabling the attendance of online classes, the study participants, above all the youngsters and their families, also reported the inadequacy of their homes as suitable places to study (for example, not having electricity, basic sanitation, or living in overcrowded environments). In addition, Ciganos/Roma children are more likely to live in overcrowded households (94%) and households without access to tap water (24%) (FRA 2022). These constraints had already been identified in international reports and studies, which, in the meantime, began to

be developed and emphasized how the pandemic crisis can disproportionally affect the education projects of young Ciganos/Roma (Bešter and Pirc 2020; FRA 2020; Korunovska and Jovanovic 2020).

> *"(...) many Ciganos/Roma families do not have access to electricity, to water, etc. and many students lost out (...) during those months, due to the fact that they don't literally have the means to learn anything."* (Focus Group 1, Young Student)

They also highlighted restrictions in access to classes that are more related to structural inequalities (CNE and CIPES 2021) than to the adverse socioeconomic conditions of these families, such as contextual difficulties of the actual schools in resuming their activities in a non-face-to-face format or in adapting assessment processes to an online format. These constraints contributed to some of the youngsters who were interviewed / ethnographically followed spending long periods of time without classes or at risk of not passing the year.

> *Due to the pandemic, João started to have classes through the Microsoft Teams online platform, but only the theoretical modules, which did not require any type of specialized software. All the modules of the technical area were suspended, with the teachers having said it was necessary to wait until the pandemic situation was over and that, until then, they should "do what they could". The youngster was worried about the possibility of all the practical assessments being postponed to the end of September, and that there would be an overload of work at that time. In contrast to João, Lucas' school decided to bring the summer holidays forward; therefore, since then, this student has not had any school task to perform. He received news from the school and knows that he will have online classes as soon as the holidays finish (he doesn't yet know when he will resume classes) through the Zoom platform, with the camera switched on, but the microphone switched off, so that only the teacher can talk. He will be submitted to an assessment system in which the teacher asks the students questions individually and they will have to switch the microphone on and answer. They will also have "worksheets" to complete, which they will receive by e-mail. However, the youngster was skeptical about this system, as the only contact his year group had attained with the school, in virtual format, up to that date, had been to receive work by e-mail and send back to the teachers. Lucas was also clearly apprehensive about the possibility of the national exams being postponed to September.* (Field Diary, Ethnographies with Young Students, April 2020)

Issues such as those indicated above, in addition to having more direct and immediate effects on access to classes, giving rise to an increase in the digital divide and the disengagement of the poorer and more vulnerable, the info-excluded (Rideout and Katz 2016; van Dijk 2005), which could also influence the engagement of young Ciganos/Roma with school in a manner that is invisible and difficult to measure. This would be reflected in an educational disconnection of young Ciganos/Roma, which brings us to a much wider motivational dimension (ERGO Network 2020) that will be, for this reason, explored below. In fact, the disaffection and disinterest shown by some students could be inherent to their individual personality, but also involves family, social and contextual dimensions related to the educational circumstances and environments provided to the students (González and Bernárdez-Gómez 2019).

### 4.1.2. Discontinuities in Daily Routines and Significant Relationships

In a broad sense, school can be understood as a space of sociability, protection, and wellbeing, but also, in the context of citizenship, a place where social skills are developed and where personal and social identities are forged (CNE and CIPES 2021; Rios and Markus 2011). Hence, it is important to stress the impacts of the pandemic related to the abrupt cutting of relevant social relationships and routines, as the dynamics of these routines outside the family, frequently interethnic, are identified as important facilitators of the educational achievement of Ciganos/Roma students (Durst and Bereményi 2021; Pantea 2015). This breakdown of relationships and routines, and consequent social isolation (from peers, but also from teachers and other school/community professionals), is particularly

emphasized by young students and their parents, and expressed as a particular form of concern.

*"(...) I am in my first year [at university], I have just finished, but I was having the best time ever, because the people there didn't judge me because of what I was, my past experiences, what I did . . . (...) nobody judged me, I was me and that was it. And I'll tell you, it was really tough for me to go home and have, for example, classes like this, because it was just at a time when they weren't judging me and were getting closer to me, shall we say, it was at that moment this happened. And at that moment when we feel that we can do something (...) And (...) in which everyone is getting closer to you (...) this happens and at that moment you feel that it's the right time, it's now that I am going to manage to change the mentality, probably not 100%, but manage ( . . . ) to make them think about it seriously (...)"* (Focus Group 1, Young Student)

*"I think that there has been a decline in my children, why? Because of their habits, because they don't have schedules, basically. They are at home, they are protecting themselves, right, they are not completely in the streets, basically they are living at home, and what happens? At home, they don't have almost anything to do, they don't feel like doing school tasks, they sleep, (stay up) until 1 or 2 in the morning (...) And at school, they had to go to school, socialize with other children, you might not think so but they had other activities, another motivation to do their work, to do better, and like this there just isn't, there isn't and this situation is more difficult."* (Focus Group 2, Mother of Young Students)

In the same way as with social relationships, the pandemic, and particularly the lockdown, came to affect the supporting role of the associativism and participation in community projects and religious activities (Gamella 2011; Gofka 2016), primarily due to the suspension of their activities and/or difficulties in their readaptation to an online format and inability to benefit from local study support projects such as the Escolhas Program (Lopes 2021). This issue was emphasized transversally by various participants in the research, as observed in the reports, and could have indirect effects, as yet difficult to measure, on the schooling of young Ciganos (Bešter and Pirc 2020).

*"(...) if it were not this situation, as a rule, what they do is go to school, then they come and study and do their homework with us, and we try to motivate them as much as possible (...). At the moment it's going to be very difficult to motivate them to go to school."* (Interview, Escolhas Project Technician, AML)

*"But now there's been (...) this outbreak, again, this new peak and we decided, for safety reasons, that it was best to close the churches again for a fortnight. Just so as to help in (...) not spreading the virus through us. (...) it has been the church, we have done (...) this mediation. Ultimately, the work of a pastor, nowadays, in the midst of society, has been mediation and it has been a very successful mediation in that regard. Of course, not all are successful, there are also some who get tired half way along the path (...) and leave, but in 90% of the cases, we are successful with this."* (Interview, Evangelical Pastor, AML)

While the socioeconomic context in which families live appears to have direct and immediate effects on access to classes online, the deterioration of the living conditions of some Ciganos/Roma families during the pandemic was also reported as a demotivation for the continuity of educational pathways, especially for youngsters whose parents essentially subsist on informal work (for example, itinerant vending and fairs) and saw their business activities interrupted, frequently leaving no social protection alternatives (ERGO Network 2020). Some of these youngsters felt, inclusively, that it was their duty to suspend their schooling and search for a job that could bring in extra income to support their family.

*"(...) there's a lot of informal work which also doesn't help these families at all because in a situation like this they are completely unprotected and then there's no form of applying for parental leave, there's no sick leave, and even if there is, it's limited. I was just with a*

*family that had a new born baby, the baby had heart problems and the mother went to hospital with the baby where she was hospitalized, because of pseudo-Covid screening the mother had to stay. And then the situation immediately started getting more complicated because the father actually works and has a contract, but the issue was that if the father stayed at home, he would automatically lose 40% of his wage (...) and this is complicated (...) in these households."* (Focus Group 3, Escolhas Project Technician, AMP)

Added to the fact of having less income, the more socioeconomically vulnerable households, which is the case of many Ciganos/Roma families, also experienced increasing family expenses during the pandemic. This aspect is particularly emphasized through the professional lens of the Escolhas Program Technicians.

*"Then we also had to support the families here, try to understand their needs in terms of foodstuffs, because it's like this, many households didn't experience a change in economic circumstances because those that are social insertion income beneficiaries stayed the same, but they had their children at home throughout the entire day, meaning more meals (...) and when they are engaged in summer or easter activities they are in the project and we provide a meal here (...)"* (Interview, Escolhas Project Technician, AMP)

Likewise, the review by Nicola et al. (2020) highlights that, particularly during the lockdown periods, these families were forced to provide meals for all the household members, which does not happen in non-pandemic scenarios due to the social support received by some children to this end as a result of participating in extracurricular activities and homework supervision, in the meantime suspended. Once again, the protective role of social and community resources that previously helped to maintain several projects of educational continuity was abruptly stopped with the pandemic, highlighting even more the social and digital gap between many Ciganos/Roma families.

## 5. Conclusions

The processes of otherness, social 'bordering', and confinement of entire Roma settlements and neighborhoods in Europe during the pandemic intensified socioeconomic inequalities (Sarafian 2022). While it is true that digital disparities can contribute to educational inequalities, these disparities cyclically reinforce and perpetuate socioeconomic inequalities. In fact, the impacts indicated by the different actors involved in the study, in a transversal and crossed perspective, point to an exacerbation of the inequalities that already existed before the pandemic, with the health situation revealing the production and reproductions of inequalities in access to education, but also in relation to the living conditions assuring subsistence (Bourdieu and Passeron 1983). In the European Union, the various lockdowns accompanied by digital learning measures left more than half the Roma children and youngsters without access to school and facilities that would enable them to receive education and attain successful educational outcomes, thus further fueling the already high school dropout rates among Roma students (Korunovska and Jovanovic 2020). This digital gap has severely affected the new Roma generations and their education learning processes, in a manner as yet unknown. Accordingly, this study constitutes a relevant starting point to empirically reflect on the direct and indirect constraints from the digital divide in the Roma/Ciganos students' access to school during the pandemic, where online learning was an unavoidable reality. Besides this, extended the existing knowledge on the different-level interrelated determinants of Ciganos/Roma educational achievement and/or continuity, at this time in the particular context of COVID-19/digital learning. Overall, this understanding may contribute to more equal practices and policies in the education field, undermining widespread forms of structural and institutional discrimination that systematically jeopardizes Ciganos/Roma families in different areas of their life (FRA 2018; Kende et al. 2021).

Indeed, the implementation of universal measures does not enable covering the population in an equal way and is clearly questionable, due to not taking into account the pre-existing structural and systemic inequalities (Magano and Mendes 2014) that were

exacerbated during the pandemic. The existing structural inequalities, including digital divides, will lead to unintended consequences. It should be stressed that these effects on the educational achievement and continuity of Ciganos/Roma children and youngsters are still enigmatic, and we have no precise understanding of their scale and intensity. Thus, we are in need of wider and longitudinal studies that may complement the presented findings.

If, on the one hand, these impacts continue to be rendered invisible and naturalized in the public sphere, on the other hand, the measures and legislation underlying the pandemic effects continue not to include Ciganos/Roma in policymaking processes. On the contrary, the participatory research approach used intended to encourage Roma/Ciganos engagement in decision-making processes regarding their education, by recognizing the youngsters'/families' own voices and those from their representative gatekeepers as robust expertise to produce knowledge with a greater potential of accurate translation into more inclusive educational practices and policies (Condon et al. 2019).

Nevertheless, this process is not without limitations. In the particular context of pandemic lockdowns, our traditional "hard to reach" population was even more difficult to approach, and it is possible that Ciganos/Roma families with serious digital constraints were not able to collaborate, precisely due to their digital divide. Overall, data collection scripts were developed previously to the COVID-19 pandemic, in the main scope of the broader project about pathways of Ciganos/Roma students attending secondary education in Portugal. Thus, specific questions about digital learning and its consequences on the youngsters' educational projects had to be introduced at the same time that the pandemic was still a new worldwide research phenomenon, which in some way may have ended up compromising the potential richness of acceded data. Even so, we believe that the presented findings, even collected in the beginning and more stressful phases of the pandemic, may mirror wider forms of social and digital inequalities historically portrayed by certain families and their students, and that is therefore essential to reveal even after the pandemic phenomenon.

**Author Contributions:** Conceptualization, S.M., M.M.M. and P.C.; methodology, S.M., S.P. and P.C.; software, S.M. and S.P.; validation, S.M., M.M.M. and O.M.; formal analysis, S.M., M.M.M., S.P., P.C. and O.M.; investigation, S.M., M.M.M., S.P., P.C. and O.M.; resources, S.M. and S.P.; writing—original draft preparation, M.M.M., S.P. and S.M.; writing—S.M., M.M.M., S.P., P.C. and O.M.; visualization, S.P. and S.M.; supervision, M.M.M.; project administration, M.M.M. All authors have read and agreed to the published version of the manuscript.

**Funding:** EduCig Project: School performance among the Roma: research and co-design project (PTDC/CED-EDG/30175/2017), funded by the Foundation for Science and Technology (FCT), Portugal.

**Institutional Review Board Statement:** The study was conducted in accordance with the Declaration of Helsinki, and approved by the Institutional Review Board (or Ethics Committee) of ISCTE-IUL.

**Informed Consent Statement:** Informed consent was obtained from all subjects involved in the study.

**Data Availability Statement:** The data supporting presented findings are available from the corresponding author, [M.M.M.], upon reasonable request.

**Conflicts of Interest:** The authors declare no conflict of interest.

## Notes

[1] We maintain the Portuguese emic term, as recognized and used by the Portuguese Ciganos themselves. In international contexts, the term can be understood as Portuguese Roma or Romani persons.

[2] EduCig Project: School performance among the Roma: research and co-design project (PTDC/CED-EDG/30175/2017), funded by the Foundation for Science and Technology (FCT).

[3] https://www.seg-social.pt/covid-19 accessed on 4 July 2022.

4     In Portugal, in 2020, Information and Communication Technology (ICT) Access and Usage by Households and Individuals was 84.5% of the national population (Information and Communication Technology (ICT))—Access to the Internet—OECD Data (oecd.org).

5     Integrated Education and Training Program (PIEF): which seeks to foster fulfillment of mandatory schooling and social inclusion.

6     Educational Territories of Priority Intervention (TEIP): a government initiative, currently implemented at 136 school groupings/ non-grouped schools that are located in economically and socially deprived territories.

7     Recognition, Validation, and Certification of Skills (RVCC) processes seek to identify knowledge and skills acquired over a lifetime, assessing them according to structured rules and procedures, and validating them through the awarding of school and/or vocational/professional certification.

8     Escolhas (Choices) is a governmental program created in 2001, with its current generation (7th) involving a total of 112 funded projects (30 in Northern Portugal). Its main objectives are the boosting of educational achievement and vocational training, the development of social and personal skills, and the monitoring of life projects of socially excluded youngsters, by fostering their individual abilities. Various project activities are carried out in the youngsters' own contexts, such as at schools and neighborhoods, with Education and Training being one of the program's five priority areas.

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
