# Peer review of "How Did the COVID-19 Pandemic and Digital Divide Impact Ciganos/Roma School Pathways?"

_socsci, doi:10.3390/socsci12020086_

Round 1

Reviewer 1 Report

The paper is well written and addresses an important issue: the impact of COVID-19 pandemic on the school pathway of Roman youth in Portugal. The authors argue persuasively that the lockdown impacted negatively on the educational participation and learning opportunities of this population.

I recommend the manuscript for publication after minor revisions.

The structure of the paper is well conceived and logical. The introduction contains the useful preview of the study. Each section contributes to building up the clear argument in a transparent manner.

The only structural unbalance I could observe is that the presentation of the data collection methods (Section 4: lines 240-424) outweighs the section presenting the results and their discussion (Section 5, lines 429-599). This could be corrected by a more succinct discussion of the methods (presenting the data collection more synthetically) which would allow more space for elaborating the analysis of the empirical results.

I also suggest considering the more detailed analysis and qualitative analysis of the material which could be elaborated in section 5. In the present form the interview fragments are used more like illustrations. A more in-depth analysis would enhance the value of the presented case materials and consolidate the empirical foundation of the study.

The concluding section (which could be 6 - not 5 as it is in the present) is rather general. Would be useful not only refer back to the literature but also to connect more closely to the previous section and highlight the value of the analysis and in this way emphasising the added value of this important study.

Author Response

First of all, we genuinely thank the reviewer for their kind feedback about our manuscript and for the opportunity to improve it. And we will be available to expand and improve what still deserves to be adjusted.

We are sorry we couldn't make the changes in track changes but all changes are marked in blue. We still tried to do that in the end, but we couldn't do it either and it generated errors in the text. But all changes are marked in blue. Accordingly, a point-by-point reply to all comments and suggestions now follows.

Reviewer 1

# Comment 1: The only structural unbalance I could observe is that the presentation of the data collection methods (Section 4: lines 240-424) outweighs the section presenting the results and their discussion (Section 5, lines 429-599). This could be corrected by a more succinct discussion of the methods (presenting the data collection more synthetically) which would allow more space for elaborating the analysis of the empirical results.

Answer: Thank you very much for this useful suggestion. A more succinct discussion of the methods was introduced, and the participants' overall information is now presented along with the respective data collection procedure. Methods and Materials' entire section comprises at this time, lines 252-397.

Comment 2: I also suggest considering the more detailed analysis and qualitative analysis of the material which could be elaborated in section 5. In the present form the interview fragments are used more like illustrations. A more in-depth analysis would enhance the value of the presented case materials and consolidate the empirical foundation of the study.

Answer: We take this comment very carefully. Accordingly, at the beginning of the Results' section (blue highlighted) an introductory explanation is now made of how the overall presented results interrelate with the already developed research and models on determinants of Roma/Ciganos educational achievement and continuity. Afterwards, several contextual elaborations were made on the illustrative presented materials (blue highlighted), such as a more precise link with participants' belonging or a deepest discussion about its potential implications.

Comment 3: The concluding section (which could be 6 - not 5 as it is in the present) is rather general. Would be useful not only refer back to the literature but also to connect more closely to the previous section and highlight the value of the analysis and in this way emphasising the added value of this important study.

Answer: Number 5 is a typing error and was corrected. Thank you for noticing this. Afterwards, and according to your suggestion, were deepest highlighted the empirical, theoretical and practical contributions of the previously detailed results (blue highlighted).

Reviewer 2 Report

This is an already good paper, so it could be taken into consideration for publication. However, authors have to improve some sections of this study as follows:

First, in the title and in the abstract (and even in certain areas throughout the paper) it should appear Roma as a word, because Roma is more used in English than Cigano. Maybe if it is used Roma/Cigano it would be great, so a better title could be ‘How did the Covid-19 pandemic and digital divide impact Portuguese Roma/Cigano school pathways?’

Second, introduction should better highlight what this paper brings new in current international studies on Roma/Cigano education and how this paper pushes forward what we already know in this literature.

Third, the literature review on Roma studies and their education  could be complemented by presenting school segregation of the Roma pupils in Central and Eastern Europe countries – see the study of Helen O’Nions (doi: 10.1080/14675980903491833) and Vera Messing’s study in European Education, 2017 (doi: 10.1080/10564934.2017.1280336). Moreover, important to be mentioned is how the Roma/Gypsy/Cigano people are stigmatized due to migration in Western Europe and this affected younger Roma as a fragmented habitus process (doi: 10.1111/1468-2427.13053) and how education can be an important tool to get Roma out from marginality (see doi: 10.1080/14702540802596608) because stigma aspects affected the Roma pupils at school too. Also, several other studies on Roma as outsiders and perceived as others could be mentioned - Powell R. and Lever J. (in journal Current Sociology, 2017) reflected on Roma as perennial outsiders in Europe, while the issue of otherness against the Roma people is frequently mentioned even in multiethnic neighborhoods (see an article of Jucu I.S. and Covaci R. in journal Identities, 2021).

Fourth, the empirical data and the method are good, but there are no limitations of the data and methods presented in the paper.

Finally, conclusions do not present the implications of this study on international level or how the results of this study brings new elements to what we know in Roma/Cigano education at international level.

Author Response

First of all, we genuinely thank the reviewer  for their kind feedback about our manuscript and for the opportunity to improve it. And we will be available to expand and improve what still deserves to be adjusted. We are sorry we couldn't make the changes in track changes but all changes are marked in blue. We still tried to do that in the end, but we couldn't do it either and it generated errors in the text. But all changes are marked in blue. Accordingly, a point-by-point reply to all comments and suggestions now follows.

Reviewer 2

Comment 4. First, in the title and in the abstract (and even in certain areas throughout the paper) it should appear Roma as a word, because Roma is more used in English than Cigano. Maybe if it is used Roma/Cigano it would be great, so a better title could be ‘How did the Covid-19 pandemic and digital divide impact Portuguese Roma/Cigano school pathways?’

Answer: As suggested, the Portuguese word “Ciganos” was replaced across the text by the embracing expression “Ciganos/Roma” (in blue).  

Comment 5: Introduction should better highlight what this paper brings new in current international studies on Roma/Cigano education and how this paper pushes forward what we already know in this literature.

Answer: Thank you for your suggestion. In the final of the Introduction was highlighted a more specific contribution from the study to the existing literature, namely its role on exploring the already known determinants of Ciganos/Roma educational achievement and continuity in the particular stressful context of pandemic and lockdowns, where online learning becomes an unavoidable day-to-day reality.

Comment 6: Third, the literature review on Roma studies and their education  could be complemented by presenting school segregation of the Roma pupils in Central and Eastern Europe countries – see the study of Helen O’Nions (doi: 10.1080/14675980903491833) and Vera Messing’s study in European Education, 2017 (doi: 10.1080/10564934.2017.1280336

Answer: Thank you very much for the pertinent suggestion. We have introduced complementary ideas to the issues of segregation in different countries in Central and Eastern Europe and in Portugal, in point/section 3. Educational situation before the pandemic (blue highlighted).

Comment 7: Moreover, important to be mentioned is how the Roma/Gypsy/Cigano people are stigmatized due to migration in Western Europe and this affected younger Roma as a fragmented habitus process (doi: 10.1111/1468-2427.13053) and how education can be an important tool to get Roma out from marginality (see doi: 10.1080/14702540802596608) because stigma aspects affected the Roma pupils at school too. Also, several other studies on Roma as outsiders and perceived as others could be mentioned - Powell R. and Lever J. (in journal Current Sociology, 2017) reflected on Roma as perennial outsiders in Europe, while the issue of otherness against the Roma people is frequently mentioned even in multiethnic neighborhoods (see an article of Jucu I.S. and Covaci R. in journal Identities, 2021).

Answer: We appreciate very much the pertinence of this comment and in that sense, we have introduced changes in point/section 3. Educational situation before the pandemic, in order to mention migratory movements in Europe, as well as, the relevance of education as a means to fight marginality, in essence the power that education can exercise, we introduced the perspectives of suggested authors(blue highlighted).

Comment 8: Fourth, the empirical data and the method are good, but there are no limitations of the data and methods presented in the paper.

Answer: Thank you very much for this useful comment. Regarding this, the last conclusion paragraph reflects now the main limitations of the work, preceded by its final remarking contributio (blue highlighted).

Comment 9: Finally, conclusions do not present the implications of this study on international level or how the results of this study brings new elements to what we know in Roma/Cigano education at international level.

Answer: According to this suggestion, were deepest highlighted the empirical, theoretical and practical contributions of the previously detailed result ((blue highlighted) and also the main advantages from the participatory research approach used (blue highlighted).

Reviewer 3 Report

My congratulations to the authors for this interesting article which makes visible a particularly vulnerable student body in normal situations and especially during the COVID-19 pandemic.

Although the theoretical part is well structured, I consider that it is perhaps a little long.

Author Response

First of all, we genuinely thank the reviewer for their kind feedback about our manuscript and for the opportunity to improve it. And we will be available to expand and improve what still deserves to be adjusted.  We are sorry we couldn't make the changes in track changes but all changes are marked in blue. We still tried to do that in the end, but we couldn't do it either and it generated errors in the text. But all changes are marked in blue. Accordingly, a point-by-point reply to all comments and suggestions now follows.

Reviewer 3

Comment 10: Although the theoretical part is well structured, I consider that it is perhaps a little long.

Answer: Thank you for notice this. Accordingly, information about determinants from Roma/Ciganos educational achievement and/or continuity are now only presented at a simplest sentence (blue highlighted) which highlights common points on the developed models and studies on this topic. More specific theoretical information is now moved to Discussion and Conclusion sections to in-depth the value of the presented materials (blue highlighted).

Round 2

Reviewer 2 Report

The paper is now much improved, so I am happy to accept it for publication. For the production stage of this paper, authors have just to attentively check all their references to be in Social Sciences journal style.